# Ewing Sarcoma as Secondary Malignant Neoplasm—Epidemiological and Clinical Analysis of an International Trial Registry

**DOI:** 10.3390/cancers14235935

**Published:** 2022-11-30

**Authors:** Stefan K. Zöllner, Katja L. Kauertz, Isabelle Kaiser, Maximilian Kerkhoff, Christiane Schaefer, Madita Tassius, Susanne Jabar, Heribert Jürgens, Ruth Ladenstein, Thomas Kühne, Lianne M. Haveman, Michael Paulussen, Andreas Ranft, Uta Dirksen

**Affiliations:** 1Pediatrics III, University Hospital Essen, 45147 Essen, Germany; 2West German Cancer Center (WTZ), University Hospital Essen, 45147 Essen, Germany; 3German Cancer Consortium (DKTK), Essen/Düsseldorf, University Hospital Essen, 45147 Essen, Germany; 4Pediatric Hematology and Oncology, University Children’s Hospital Münster, 48149 Münster, Germany; 5St. Anna Children’s Cancer Research Institute and Medical University Vienna, 1090 Vienna, Austria; 6Department of Oncology and Haematology, University Children’s Hospital Basel, 4056 Basel, Switzerland; 7Princess Máxima Center for Pediatric Oncology, 3584 Utrecht, The Netherlands; 8Pediatric Hematology and Oncology, Vestische Children’s and Youth Hospital Datteln, University Witten/Herdecke, 45711 Datteln, Germany

**Keywords:** childhood cancer, cancer survivor, Ewing sarcoma, secondary malignant neoplasms, secondary malignancy

## Abstract

**Simple Summary:**

Ewing sarcoma (EwS) is a malignant bone and soft-tissue cancer that primarily affects adolescents and young adults. In rare cases, EwS develops as a secondary cancer; that is, after a previous cancer other than EwS. We collected information on all patients with EwS as a secondary cancer from three past international EwS studies to better understand affected patients and to identify potential at-risk patients. Forty-two patients with secondary EwS were identified, representing approximately 1.1% of all EwS cases. As primary cancers, patients suffered mainly from cancers of the blood-forming system, such as leukemia and lymphoma. We could not identify any risk factors for the development of EwS as a secondary cancer. Survival from a second cancer diagnosis with EwS is comparable to EwS as a first cancer diagnosis; therefore, patients with secondary EwS should also be offered complete therapy with the goal of cure, especially if the tumor is localized to only one site.

**Abstract:**

Ewing sarcoma (EwS) is the second most common bone and soft tissue tumor, affecting primarily adolescents and young adults. Patients with secondary EwS are excluded from risk stratification in several studies and therefore do not benefit from new therapies. More knowledge about patients with EwS as secondary malignant neoplasms (SMN) is needed to identify at-risk patients and adapt follow-up strategies. Epidemiology, clinical characteristics, and survival analyses of EwS as SMN were analyzed in 3844 patients treated in the last three consecutive international EwS trials, EICESS 92, Euro-E.W.I.N.G. 99, and EWING 2008. Forty-two cases of EwS as SMN (approximately 1.1% of all patients) were reported, preceded by a heterogeneous group of malignancies, mainly acute lymphoblastic leukemias (*n* = 7) and lymphomas (*n* = 7). Three cases of EwS as SMN occurred in the presumed radiation field of the primary tumor. The median age at diagnosis of EwS as SMN was 19.4 years (range, 5.9–72) compared with 10.8 years (range, 0.9–51.2) for primary EwS. The median interval between first malignancy and EwS diagnosis was 7.4 years. The 3-year overall survival (OS)/event-free survival (EFS) was 0.69 (SE = 0.09)/0.53 (SE = 0.10) for localized patients and 0.36 (SE = 0.13)/0.29 (SE = 0.12) for metastatic patients (OS: *p* = 0.02; EFS: *p* = 0.03). Survival in patients with EwS as SMN did not differ between hematologic or solid primary malignancies. EwS as SMN is rare; however, survival is similar to that of primary EwS, and its risk-adjusted treatment should be curative, especially in localized patients.

## 1. Introduction

Overall survival rates for many pediatric cancers are improving decade by decade. In Europe, the 5-year survival rate for children diagnosed with malignant neoplasms reached 79.1% between 2000 and 2007, largely due to new advances in therapy and supportive care [1]. Currently, there are more than 500,000 survivors of childhood cancer in Europe [2]. As this trend continues, minimizing acute and long-term toxicity associated with treatment is of paramount importance, because most children with cancer continue to suffer from significant treatment-related toxicities [3]. Secondary malignant neoplasms (SMN) are recognized as late sequelae of cancer therapy in children [4]. SMN represent subsequent, distinct tumor entities in the same patient. They are histologically distinct from both the primary tumor and metastases from the primary tumor and usually occur within 30 years of the diagnosis of a first malignancy [5]. The marked differences in occurrence and the role of specific therapeutic exposures have led to the classification of SMN into two distinct categories: radiation-induced solid SMN and chemotherapy-induced myelodysplastic syndrome (MDS)/acute myeloid leukemia (AML) [6].

Ewing sarcoma (EwS) is a rare but highly aggressive bone and soft-tissue malignancy that usually affects adolescents and young adults [7]. EwS is one of the small blue round-cell sarcomas characterized by chromosomal translocations in which ETS transcription factors are fused to a member of the FET gene family [8]. The most common tumor-specific chimeric transcription factor, EWSR1-FLI1, consists of the Ewing sarcoma breakpoint region 1 protein (EWSR1) and an ETS family gene such as the Friend leukemia integration 1 transcription factor (FLI1) [7,9]. Therapies targeting EWSR1-FLI1 would provide a tumor-specific targeted approach but have yet to be established in routine clinical practice. Currently, the standard treatment for EwS patients is multimodal and includes intensive polychemotherapy, as well as surgery and/or radiotherapy for local therapy [10].

Previous studies have described a greater than ninefold increased risk of developing sarcoma in childhood cancer survivors compared with the general population [11]. While EwS survivors are at increased risk for SMN compared with the age- and sex-matched general population, EwS itself may also present as SMN [12,13]. EwS as SMN accounts for a minority of all EwS cases, for which limited data are available [13,14,15,16]. Despite its rare occurrence, patients with EwS as SMN appear to have clinical differences and a worse prognosis than patients with primary EwS [17].

To supplement data on this patient group, we retrospectively analyzed the epidemiology and clinical features of patients with EwS as SMN over a 27-year period who were enrolled in EwS trials and registries. The aim of this study was to retrospectively describe the incidence, epidemiology, and clinical characteristics, as well as the risk factors of patients with EwS as SMN. In one of the largest clinical studies on this topic, we have identified forty-two patients with EwS as SMN, representing approximately 1.1% of all reported EwS cases. The present study suggests that the outcome of patients with EwS as SMN is determined by the staging at diagnosis and not by the diagnosis of a secondary malignancy per se [4]. Survival in EwS as SMN does not differ between hematologic or solid primary malignancies, and risk factors for EwS as SMN appear to be similar to risk factors of other non-EwS SMN [18].

## 2. Materials and Methods

### 2.1. Patient Cohorts and Eligibility Criteria

From 1991 to 2019, three consecutive and multinational EwS trials, namely the European Intergroup Cooperative Ewing’s Sarcoma Studies EICESS 92 (ClinicalTrials.gov identifier: NCT00002516), the European Ewing Tumor Working Initiative of National Groups 1999 Euro-E.W.I.N.G. 99 (ClinicalTrials.gov identifier: NCT00020566), and the EWING 2008 (ClinicalTrials.gov identifier: NCT00987636), enrolled 3844 patients with primary EwS and EwS as SMN. All trial protocols were conducted in accordance with the ethical principles of the Declaration of Helsinki and the guidelines for Good Clinical Practice, reviewed by the appropriate institutional review boards, and approved by an independent ethics committee.

Several trials aimed to optimize treatment and outcome for EwS patients. Because primary malignancy was an exclusion criterion for randomization, patients with EwS as SMN were not eligible for randomization and were treated according to standard protocols. Patients with EwS as SMN were included as registry patients in the EWING registry of the GPOH (German Society of Pediatric Hematology and Oncology), which allowed analysis of data on patients not registered as trial patients. The GPOH database was last updated in June 2019.

The differences in risk stratification and treatment strategies across the different Cooperative Ewing Sarcoma Study (CESS)-led EwS trials is shown in Table 1. In the EICESS 92 protocol, patients with localized tumors and a tumor volume of <100 mL were classified into a standard-risk (SR) stratification, in contrast with high-risk (HR) patients who presented with metastasis or a tumor volume of ≥100 mL at diagnosis. Treatment consisted of vincristine, actinomycin D, ifosfamide, and adriamycin (VAIA) for standard-risk patients and VAIA plus etoposide (EVAIA) for high-risk patients. Local control included surgery, radiotherapy, or a combination of both. In the Euro-E.W.I.N.G. 99 trial, four different risk groups were defined, namely R1, R2pulm, R2loc, and R3. Standard-risk patients assigned to R1 either had a good histologic response to neoadjuvant treatment (<10% viable cells), a tumor with a baseline volume of <200 mL and after resection at diagnosis, or were treated with radiotherapy alone as local treatment. Patients with localized disease and either a poor histologic response (≥10% viable cells) to induction chemotherapy or with a tumor volume of ≥200 mL who were either unresected at diagnosis or initially resected were included in R2. Patients who had disseminated disease at diagnosis were assigned to R3. At the start of the Euro-E.W.I.N.G. 99 trial, treatment regimens were further stratified for patients with isolated pulmonary metastasis who were assigned to R2pulm. In induction chemotherapy, patients received 6 cycles of VIDE followed by treatment with vincristine, actinomycin D, Ifosfamide (VAI), and for local therapy, either surgery, definitive irradiation, or surgery and irradiation combined. In addition, R1 patients received 7 courses of vincristine, actinomycin D, cyclophosphamide (VAC) or 7 courses of vincristine, actinomycin D, ifosfamide, adriamycin (VAI) after induction chemotherapy. R2loc patients received 7 courses of VAI or 1 course of VAI and high-dose chemotherapy with busulfan and melphalan (Bu-Mel), whereas R2pulm patients received all 7 courses of VAI consolidation chemotherapy followed by lung irradiation or Bu-Mel. The EWING 2008 trial was a joint protocol of international EwS trial groups. Patients were stratified into R1 patients with good histologic response (<10% viable tumor cells) and localized disease, R2loc patients with poor histologic response (≥10% viable tumor cells) and localized disease, R2pulm patients with exclusive lung metastases, and R3 patients with disseminated disease at diagnosis, i.e., lung metastases and metastases to other sites (Table 1).

The GPOH database of these three studies was screened for EwS as SMN. Patients were identified based on the following criteria: First malignancy in all age groups, histologically confirmed diagnosis of EwS as SMN (non-EwS round cell sarcoma were excluded), treated analogously to the EICESS 92, Euro-E.W.I.N.G. 99, and EWING 2008 trial protocols. Patients with EwS as a third or fourth malignancy were not included in this study.

### 2.2. Statistical Analysis

Descriptive statistics were calculated using the SPSS Statistics 29 (IBM Corporation, Armonk, NY), SAS 9.4 (SAS Institute, Cary, NC, USA), and MEDCALC software packages. Overall survival (OS) and event-free survival (EFS) were calculated using the Kaplan–Meier method. Univariate comparisons were estimated using the log-rank test. Fischer’s exact test was used in the analysis of contingency tables.

## 3. Results

In this retrospective analysis of the EWING registry of the GPOH, a total of 3844 patients had been enrolled in the EICESS 92, the Euro-E.W.I.N.G. 99, and the EWING 2008 trials. We identified forty-two patients with EwS as SMN with a median follow-up of 2.6 years, ranging from 0.1 to 14.6 years. According to the different trials, 8 (0.9%), 16 (1.0%), and 18 (1.3%) patients in EICESS 92, Euro-E.W.I.N.G. 99, and EWING 2008, respectively, had EwS as SMN.

### 3.1. Epidemiology of Primary Malignancies

EwS as SMN was preceded by a heterogeneous group of primary malignancies. Sixteen patients were diagnosed with neoplasms of the hematopoietic and lymphoid tissue as first malignancy. Acute lymphoblastic leukemia (ALL) was present in almost half of these cases (*n* = 7). Seven cases of EwS as SMN occurred after Hodgkin lymphoma (*n* = 4) and non-Hodgkin lymphoma (*n* = 3). Four cases of breast cancer and two cases each of osteosarcoma, retinoblastoma, melanoma, and rhabdomyosarcoma were noted. One patient was initially diagnosed with Langerhans cell histiocytosis. One case each of nephroblastoma, hepatoblastoma, malignant hemangiopericytoma, testicular teratoma, renal cell carcinoma, cervical cancer, and ovarian germ cell tumor was reported. Only one patient was diagnosed with Li-Fraumeni syndrome, a hereditary cancer predisposition syndrome [19]. In this patient, the first malignancy was a mediastinal rhabdoid tumor and the patient developed EwS as SMN seven years later. Figure 1 shows the localization of the first malignancy in relation to the location of EwS as SMN.

### 3.2. Patient Characteristics and Clinical Features of Ewing Sarcoma as Secondary Malignant Neoplasms

The clinical features and patient characteristics of EwS as SMN cases in the different trials are summarized in both Table 2 and Table 3. Table 2 presents the clinical characteristics of patients with EwS as SMN vs. EwS as primary malignancy. The gender distribution of EwS as SMN was balanced, with slightly more female (52.4%) than male (47.6%) patients. The majority of patients were older than 14 years (66.7%) at the time of EwS diagnosis. The median age at diagnosis of the first malignancy was 10.8 years (range 0.9 to 51.2), whereas EwS as SMN occurred at a median age of 19.4 years, with a range of 5.9 to 72 years. The median interval between the diagnosis of the first malignancy and the diagnosis of EwS as SMN was 7.4 years, whereas the first cases of EwS as SMN occurred as early as 1 year after the primary tumor. The longest interval between primary EwS and EwS as SMN was 41.4 years. There was no significant difference between the type of primary tumor of hematologic or solid origin with respect to the latency to EwS as SMN (*p* = 0.55; Figure 2).

Comparison between EwS as primary malignany and EwS as SMN did not reveal statistically significant differences in clinical features at diagnosis and response to treatment after induction therapy, except for tumor localization at diagnosis (*p* = 0.03; Table 2). We observed fewer EwS as SMN in the lower extremities and more thoracic and extraosseous EwS as SMN.

EwS as SMN occurred most frequently in the thorax (26.8%), pelvis (22%), extraosseous sites including the abdomen (22%), and both the cranium and lower extremities (9.75% each) (Figure 1 and Table 2). Localized disease was present in 28 (67%) patients, 4/8 in the EICESS 92 trial, 10/16 in the Euro-E.W.I.N.G. 99 trial, and 14/18 in the EWING 2008 trial. Accordingly, 14 (33%) patients with EwS as SMN had metastatic disease at diagnosis. According to the Euro-E.W.I.N.G. 99 trial, risk stratification strategies distinguished between exclusive lung metastases and disseminated disease with lung metastases and metastases to other sites. In this context, six patients were diagnosed with disseminated EwS as SMN, two of whom had exclusive lung metastases in the Euro-E.W.I.N.G. 99 trial. In the EWING 2008 trial, one patient was diagnosed with disseminated disease without lung metastases, and three patients had isolated lung metastases (Table 2). Overall, disease stage did not differ statistically between primary tumor type and EwS as SMN patients (*p* = 0.25). Notably, six patients with disseminated EwS disease had a solid primary tumor, but only one patient with disseminated disease had ALL as an initial tumor. The tumor status of patients with EwS as SMN at the time of diagnosis is shown in Table 3. The majority of patients (60.5%) had a tumor volume < 200 mL at diagnosis (Table 2).

### 3.3. Treatment Management of Primary Malignancies

Despite the heterogeneity of the first malignancies, therapy of the primary diseases followed GPOH- and non-GPOH-standardized national trials, e.g., ALL/NHL-BFM-86 (ALL), ALL-BFM 95 (ALL), COSS-86 (osteosarcoma), MAKEI 96 (germ cell tumor), or CWS-86 (soft-tissue sarcoma; EwS excluded). Across trials, nine patients (33%) received radiotherapy for the primary tumor before developing EwS as SMN. In three of the nine irradiated patients, EwS as SMN occurred near the site where the primary malignancy had been localized and irradiated (Figure 1).

### 3.4. Treatment Management of Ewing Sarcoma as Secondary Malignant Neoplasms

Treatment of EwS followed the specific trial protocols. In the EICESS 92 trial, primary malignancies were an exclusion criterion for randomization, and patients with EwS as SMN received standard therapy. In contrast, patients included in the Euro-E.W.I.N.G. 99 protocol were not excluded from stratification and received treatment according to the standard arms of the trial. In the EWING 2008 trial, patients with EwS as SMN received similar treatment, with zoledronic acid introduced in R1 patients and high-dose treosulfan and melphalan (Treo-Mel) chemotherapy in R3 patients. Further explanation of the trial design and risk strata can be found in the “Materials and Methods” section.

### 3.5. Outcome of Patients with Ewing Sarcoma as Secondary Malignant Neoplasms

Data on follow-up of patients with EwS as SMN were available from 41 patients with a median follow-up of 2.6 years (range 0.1–14.6). During the 27-year follow-up period, the 3-year overall survival (OS) was 0.69 (SE = 0.09) for patients with localized disease and 0.36 (SE = 0.13) for patients with metastatic disease (*p* = 0.02; Figure 3). The 3-year event-free survival (EFS) was 0.53 (SE = 0.10) for patients with localized disease and 0.29 (SE = 0.12) for patients with metastatic disease (*p* = 0.03; Figure 3). There was no significant difference in OS or EFS of patients with EwS as SMN between primary hematologic or primary solid malignancy (*p* > 0.05; Figure 4). Statistically, due to the small number of cases, a comparison of EFS/OS between the different EwS studies is not meaningful, partly because localized patients are mixed with metastatic patients. A similar limitation would have been the analysis of other prognostic factors known for primary EwS, such as tumor volume and histopathologic response to therapy.

## 4. Discussion

Cancer survivors are at high risk of developing long-term complications such as secondary malignant neoplasms (SMN) [20]. In childhood cancer survivors, the cumulative risk of developing a SMN 20 years after the primary diagnosis can be as high as 12% [21]. According to the German Childhood Cancer Registry, 6.8% (more than 1500 patients) of all German childhood cancer survivors diagnosed under the age of 15 were diagnosed with a SMN within 30 years of their first diagnosis between 2009 and 2018 [22].

EwS is a rare but highly aggressive bone and soft-tissue tumor [23]. Data on treatment management and survival of patients with EwS as SMN are conceivably limited [24]. In the present study, we analyzed the patient characteristics and survival data of patients with EwS as SMN in a heterogeneous group of patients with primary malignancies. EwS as SMN accounted for approximately 1.1% of all 3844 EwS cases treated between 1991 and 2019 in three consecutive and international EwS trials, i.e., EICESS 92, Euro-E.W.I.N.G. 99 and EWING 2008. The incidence of EwS as SMN is presumably lower than that of other solid bone and soft-tissue tumors such as rhabdomyosarcomas and osteosarcomas [25,26]. The median age of 19.4 years at diagnosis of EwS as SMN in our cohort is comparable to the general peak incidence age for primary EwS of 15 years [7]. While most SMN develop within 10 years, the incidence of SMN in survivors of childhood cancer increases with age, with a cumulative incidence of more than 20% 30 years after diagnosis of the primary cancer [4]. In one study, sarcomas occurred a median of 11.8 years after the diagnosis for primary malignancy [11]. In comparison, the median time from primary malignancy to EwS as SMN in our study was 7.4 years, which is higher than latency times of 3.3 years to be calculated in other EwS-specific publications (6.9 years after primary hematologic malignancies, reviewed in [15]). There was no significant difference between the types of primary tumor of hematologic or solid origin regarding the timing of development of EwS as SMN (Figure 2). The short latency period compared with other secondary soft-tissue tumors supports close follow-up after completion of treatment for a primary malignancy [18,27]. However, given the low incidence of EwS as SMN, there is no reason to perform standardized radiologic screenings apart from routine follow-up. Alternative biomarkers such as liquid biopsies that can detect EwS at an early stage are not currently available.

In a previous study of 58 patients with EwS as SMN, affected patients showed poorer survival (34.3% vs. 52.2%), smaller tumor volumes (75.0% vs. 48.2% <8 cm), and more axial tumors (77.4% vs. 62.5%) compared with patients with primary EwS [17]. Our analysis showed that the initial tumor volume and the frequency and distribution of distant metastases at the time of diagnosis in EwS as SMN is similar to that in primary EwS (Table 2) [7]. The difference in tumor location in our study between primary and secondary EwS tumors must be critically questioned due to the low case number for EwS as SMN cases. Overall, survival rates in patients with both primary EwS and EwS as SMN are comparable, although this observation may be influenced again by the small number of cases in the present study. In a large previous study of EwS and PNET as SMN cases, these patients showed worse overall survival than primary EwS patients, and this was related to older age with comorbidity [17]. That we no longer saw this association may be due to the newer data and improved treatment options, including improved supportive care. Our prognostic analysis suggests that the outcome of patients with EwS as SMN is determined by the greatest risk factor of EwS, the spread of the disease at diagnosis, and not by the diagnosis of secondary malignancy per se; this is comparable to the results of previous studies [4,17]. As the most important unfavorable prognostic factor in both primary EwS and EwS as SMN is the presence of distant metastases at the time of diagnosis [28,29], staging remains crucial in the diagnosis of EwS as SMN. Although patients with EwS as SMN have similar outcomes compared with primary EwS patients, EwS as SMN are not eligible for randomization in either the current international Cooperative Ewing Sarcoma Study (CESS) registry or the upcoming CESS trial and are thus excluded from potential new therapeutics.

In our study, the primary malignancies that preceded EwS had a wide spectrum of diseases. Most patients had primary malignancies of the hematopoietic and lymphoid tissues. ALL was the most diagnosed primary malignancy. This is not surprising, as leukemia and lymphomas are the most common malignancies in children [30]. The 25-year cumulative incidence of a secondary malignancy in children surviving ALL was estimated to be 5.2% based on the Childhood Cancer Survivor Study (CCSS) [31]. In comparison, the same CCSS analysis showed a 20-year cumulative incidence of 1.7% for all secondary malignant neoplasms in survivors of childhood AML [32]. The frequency of hematologic malignancies preceding EwS as SMN in our study is at odds with previous findings that patients with a primary sarcoma were more likely to develop a secondary sarcoma [11]. The type of primary malignancy did not correlate with disease status or with the type of metastasis of EwS as SMN, and overall did not affect the prognostic outcome of patients with EwS as SMN (Table 3, Figure 4).

Given that patients with the most common primary malignancies in our cohort, ALL and lymphoma (11/24), were most likely to receive multichemotherapy with the alkylating agent cyclophosphamide and the topoisomerase inhibitor etoposide, and given previously published data on the risk of these agents to develop SMN [33,34,35], it is possible that these agents predispose for EwS as SMN among other SMN. Because of the heterogeneity of treatment protocols for primary malignancies, variable doses of chemotherapeutic agents, lack of clinical data on patients, and small EwS case series, the specific risk of these chemotherapeutic agents for EwS as SMN cannot be conclusively assessed. Controversy surrounds the effect of radiation: for ALL survivors, irradiation is thought to be responsible for most SMNs [36]. Two previous reports described an increased risk of secondary sarcomas in patients who received radiotherapy during treatment for their first malignancy [11,17]. In the present study, EwS as SMN occurred within the presumed former irradiation field in one-third of initially irradiated patients (Figure 1). Previous studies correlated EwS as SMN with the location of the previous radiation field in 12.1% of patients [17], while individual case reports may well show higher rates without accounting for publication bias [15]. While most EwS as SMN cases are likely due to established clinical risk factors for secondary malignancies, such as irradiation and/or chemotherapy, few cases may have a biological predisposition in the form of underlying mutations. There are several genetic syndromes with links to soft-tissue and bone sarcomas, whereas EwS does not appear to be an associated index tumor, unlike osteosarcoma and embryonal rhabdomyosarcoma [37]. In EwS, a balanced chromosomal translocation t(11;22)(q12;q24) leading to the specific fusion protein EWSR1–FLI1 appears to be genetically responsible for the tumor [38]. Apart from this translocation, other genetic alterations or mutations are rare [7]. Mutations of the p53 tumor suppressor gene are detected in 5–7% of EwS cases [39]. In our study, we identified only one case of EwS as SMN with an underlying Li-Fraumeni syndrome characterized by germline mutations of the tumor-suppressor gene p53 [40]. It must be taken into account that we did not have information on germline mutations in most patients with EwS as SMN. We tried to retrospectively examine the tumor samples available to us with regard to P53 expression, but the amount of available tumor material, as well as the interpretation, did not allow any clear conclusions, so this information was not explicitly included in the study. In conclusion, it remains unclear to what extent the type of treatment of the first malignancy or a genetic predisposition contributed to the development of EwS as SMN cases, or whether these cases were incidental findings.

## 5. Conclusions

In recent years, therapy management has been modified to achieve a better balance between acute therapy-related toxicity, long-term sequelae, and efficacy to prolong survival. The successes in treating children with cancer should not be overshadowed by the occurrence of secondary malignancies, but patients and healthcare providers need to be aware of risk factors for secondary malignancies, including sarcomas, so that surveillance is targeted, and early prevention strategies are implemented. EwS as SMN are rare and occur within 10 years of primary diagnosis. Overall, clinical surveillance is complicated by the lack of clear clinical features in primary disease or treatment that precede the development of EwS as SMN.

The survival rate of EwS as SMN is comparable to that of primary EwS, and patients with EwS as SMN should be treated with curative intent.

## Figures and Tables

**Figure 1 cancers-14-05935-f001:**
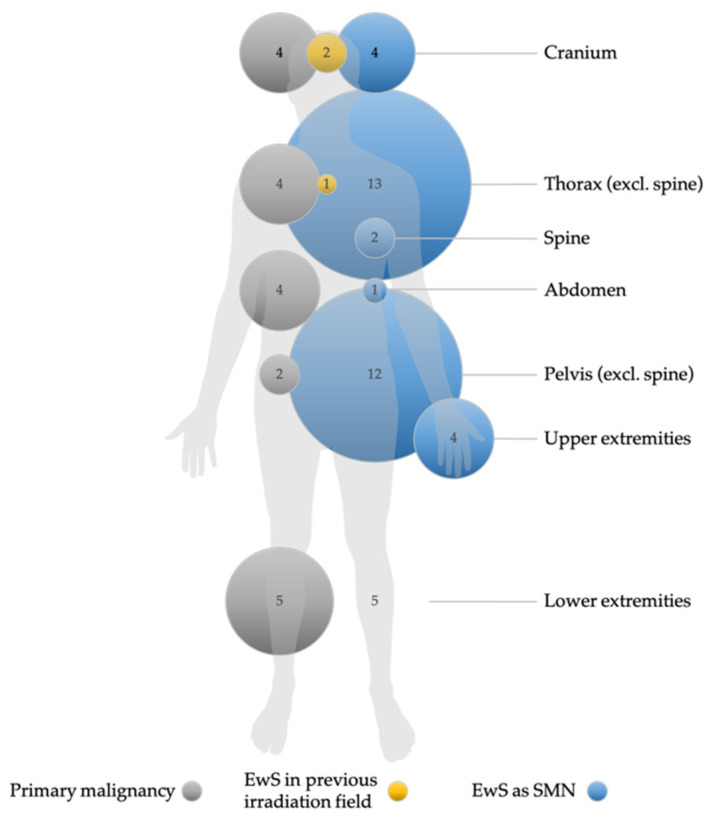
Localization and frequency of primary malignancies and Ewing sarcoma as secondary malignant neoplams (EwS as SMN), and occurrence of EwS as SMN in previously irradiated fields. Non-EwS malignancies are shown with gray circles, EwS cases with blue circles, and cases of EwS as SMN in previously irradiated fields with yellow circles. The numbers indicate the number of tumors per localization. The size of the circles is proportional to the number of cases. Missing circles mean no tumor case in this localization. Data for primary malignancies, EwS as SMN, and EwS as SMN in previously irradiated localizations (i.e., information on treatment modalities) were available in *n* = 36, *n* = 41, and *n* = 27, respectively. Nine patients were irradiated in the primary tumor setting.

**Figure 2 cancers-14-05935-f002:**
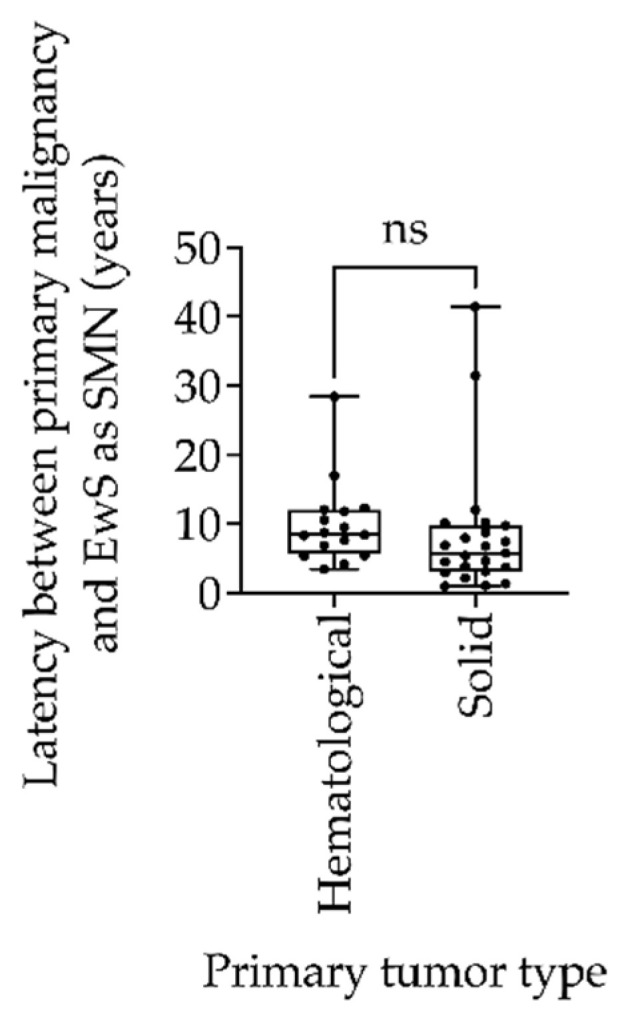
Box-and-whisker plot of latency between primary tumor and Ewing sarcoma as secondary malignant neoplasms (EwS as SMN) compared between hematologic and solid primary tumors in the EICESS 92, Euro-E.W.I.N.G. 99, and EWING 2008 trials. Statistical analysis by unpaired *t* test, ns = non-significant (*p* = 0.55).

**Figure 3 cancers-14-05935-f003:**
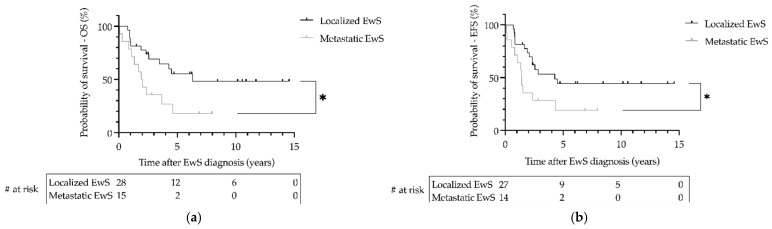
Survival of patients withEwing sarcoma (EwS) as secondary malignant neoplasms as a function of metastatic status at diagnosis in the EICESS 92, Euro-E.W.I.N.G. 99, and EWING 2008 trials. (**a**) Overall survival (OS) and (**b**) event-free survival (EFS) were calculated using the Kaplan–Meier method. * (**a**) *p* = 0.02, (**b**) *p* = 0.03. # at risk—numbers at risk.

**Figure 4 cancers-14-05935-f004:**
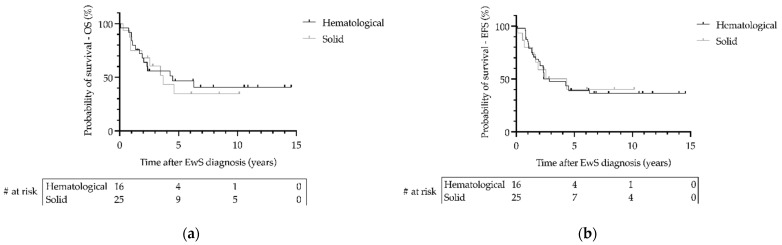
Survival of patients with Ewing sarcoma (EwS) as secondary malignant neoplasms as a function of type of primary hematological or primary solid malignancy in the EICESS 92, Euro-E.W.I.N.G. 99, and EWING 2008 trials. (**a**) Overall survival (OS) and (**b**) event-free survival (EFS) were calculated using the Kaplan–Meier method. Statistical analysis by Log-rank test. (**a**,**b**) *p* > 0.05. # at risk—numbers at risk.

**Table 1 cancers-14-05935-t001:** Development of risk stratification and chemotherapy for Ewing sarcoma (EwS) in different trials from the Cooperative Ewing Sarcoma Study (CESS) group. SR—standard-risk, HR—high-risk, VHR—very high-risk. Please see “Materials and Methods” section for more detailed information on risk strata, including randomization of R1/R2/R3. Vol.—tumor volume at diagnosis, pulmonary mets—exclusive lung metastases at diagnosis. V—vincristine, A—actinomycin D, C—cyclophosphamide, I—ifosfamide, D—doxorubicin (adriamycin), E—etoposide, BU—busulfan, ME—melphalan, etoposide, MEL—melphalan, TREO—treosulfan.

EwS Trial	EICESS 92				EURO E.W.I.N.G. 99								EWING 2008					
Number of cycles	14				14		14	8	8				14		14	8	14	15
Risk strata	SR		HR		R1 = SR		R2 = HR		R3 = VHR				R1 = SR		R2 = HR		R3 = VHR	
Clinical criteria for risk strata	localized vol. < 100 mL		metastasized vol ≥ 100 mL		localized vol. ≥ 200 mL <10% viable cells *		locolazied/ pulmonary mets vol. ≥ 200 mL ≥10% viable cells *		metastasized				localized <10% viable cells *		localized/ pulmonary mets ≥10% viable cells *		metastasized	
Chemotherapeutic regimen	VAIA + VACA	VAIA	VAIA	EVAIA	VIDE + VAC ♀	VIDE + VAI ♂	VIDE + VAI	VIDE + VAI + BU/MEL	VIDE + VAI + ME/ME	VIDE + VAI + TREO/MEL	VIDE + VAI + BU/MEL	VIDE + VAI	VIDE + VAC ♀	VIDE + VAI ♂	VIDE + VAI	VIDE + VAI + BU/MEL	VIDE + VAC	VIDE + VAC + TREO/MEL

* viable tumor cells in histological tumor resection specimen.

**Table 2 cancers-14-05935-t002:** Clinical features at diagnosis and response to therapy in comparison between Ewing sarcomas (EwS) as primary malignancy and EwS as secondary malignant neoplasms (SMN) in the EICESS 92, Euro-E.W.I.N.G. 99, and EWING 2008 trials. Fisher’s exact test was used in the analysis of contingency tables.

Clinical Features at Diagnosis (Avaiblabe Data for EwS as SMN)		EICESS 92, Euro-E.W.I.N.G. 99, EWING 2008		*p* Value
		EwS as Primary Malignancy (%)	EwS as SMN (%)	
Gender (*n* = 42)	Male	2238 (58.9)	20 (47.6)	0.16
	Female	1564 (41.1)	22 (52.4)	
Age (*n* = 42)	<14 years	1609 (42.3)	14 (33.3)	0.27
	≥14 years	2193 (57.7)	28 (66.7)	
Tumor stage (*n* = 42)	Localized	2496 (67.1)	28 (66.7)	1.00
	Metastasized	1223 (32.9)	14 (33.3)	
Tumor localization (*n* = 41) ^#^	Cranium	152 (4.1)	4 (9.75)	0.03
	Thorax (excl. spine)	628 (16.9)	11 (26.8)	
	Spine	252 (6.8)	1 (2.4)	
	Extraosseous ^#^ (incl. abdomen)	552 (14.9)	9 (22.0)	
	Pelvis (excl. spine)	841 (22.6)	9 (22.0)	
	Upper extremities	262 (7.0)	3 (7.3)	
	Lower extremities	1030 (27.7)	4 (9.75)	
Tumor volume (*n* = 38)	<200 mL	2000 (57.7)	23 (60.5)	0.75
	≥200 mL	1468 (42.3)	15 (39.5)	
Response assessment				
Histological regression (Salzer-Kuntschik *) (*n* = 18)	<10%	1375 (71.9)	10 (55.6)	0.19
	≥10%	538 (28.1)	8 (44.4)	

* viable tumor cells in histological tumor resection specimen ^#^ for adequate statistical analysis and comparison between EwS as first and only tumor and EwS as second malignancy, tumor location categories were adjusted and all non-osseous tumors were combined.

**Table 3 cancers-14-05935-t003:** Patient characteristics and clinical features of each patient with Ewing sarcoma as secondary malignant neoplasms in the EICESS 92, Euro-E.W.I.N.G. 99, and EWING 2008 trials. SR—standard-risk. Please see “Materials and Methods” section for more detailed information on risk strata, including randomization of R1/R2/R3. ALL—acute lymphoblastic leukemia. Outcome: 1 dead, 0 alive.

Trial	Patient (*n*)	Primary Malignancy	Latency (Years)	Risk Strata	Follow-Up (Years)	Outcome
EICESS 92	1	Retinoblastoma	12	R3	2.4	1
	2	Lymphoma	7.3	R2p	4.1	1
	3	Melanoma	3.1	SR	10.9	0
	4	Testicular teratoma	6.8	R3	1.7	1
	5	Cervix carcinoma	21	SR	4.3	1
	6 ^#^	Rhabdoid tumor	7.4	R2p	2.0	1
	7	Malignant hemangiopericytoma	1.3	SR	4.5	1
	8	Osteosarcoma	2.9	SR	14	0
Euro-E.W.I.N.G. 99	9	ALL	11.8	R2pulm	3.68	1
	10	ALL	12.3	R2loc	0.93	1
	11	ALL	6.8	R1	8.45	0
	12	Retinoblastoma	6.2	R3	1.92	1
	13	ALL	8.3	R1	10.15	0
	14	Non-Hodgkin lymphoma	8.75	R1	14.6	0
	15	Ovarian germ cell tumor	3.8	R1	10.6	0
	16	ALL	5.4	R1	6.1	0
	17	Nephroblastoma	4.5	R1	11.7	0
	18	Hepatoblastoma	6.9	R3	1.1	1
	19	ALL	8.7	R3	0.9	1
	20	Hodgkin lymphoma	17.0	R2pulm	0.3	1
	21	Hodgkin lymphoma	8.4	R1	3.5	1
	22	Renal cell carcinoma	1.0	R3	1.3	1
	23	Hodgkin lymphoma	28.4	R1	2.6	1
	24	Langerhans cell histiocytosis	12.0	R1	1.9	1
EWING 2008	25	Neuroblastoma	4.6	R2loc	2.33	1
	26	Osteosarcoma	7.9	R2loc	1.0	1
	27	Breast Cancer	5.4	R3	8.0	0
	28	Sweat gland carcinoma	10.1	R2pulm	6.9	0
	29	Synovial sarcoma	31.5	R2loc	1.0	1
	30	Breast cancer	3.8	R1	6.2	0
	31	Rhabdomyosarcoma	10.2	R1	6.3	1
	32	ALL	10.6	R1	6.1	0
	33	Seminoma	41.4	R2pulm	0.1	1
	34	Chronic myeloid leukemia	3.5	R1	0.9	1
	35	Rhabdomyosarcoma	5.8	R2loc	4.7	0
	36	Non-Hodgkin lymphoma	4.1	R2pulm	2.8	0
	37	Non-Hodgkin lymphoma	5.3	R1	2.4	0
	38	Hepatocellular carcinoma	9.7	NA	2.5	0
	39	Breast cancer	2.1	NA	2.4	0
	40	Hodgkin lymphoma	9.5	R1	1.5	0
	41	Liposarcoma	1.1	NA	0.8	1
	42	Breast cancer	NA	NA	NA	0

^#^ patient with Li-Fraumeni syndrome.

## Data Availability

The data presented in this study are available on request from the corresponding author.

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
