# Peer review of "Ewing Sarcoma as Secondary Malignant Neoplasm—Epidemiological and Clinical Analysis of an International Trial Registry"

_cancers, 2022, doi:10.3390/cancers14235935_

Round 1
Reviewer 1 Report
The topic is interesting but the study is poorly designed and can be misleading.
The Authors evaluated 40 patients affected by secondary Ewing sarcoma.
What about non Ewing round cell sarcoma?
Considering the very high percentage of pediatric patients affected by malignancies, 40 cases of EWS seems to be by chance rather than any connection.
That's probably why prognosis is the same as EWS as first tumor. For example, if a person had a previous car accident, the chance of having a second one is the same. He doesn't risk more the second time.
Therefore, the message is misleading.
Author Response
The topic is interesting but the study is poorly designed and can be misleading.
The Authors evaluated 40 patients affected by secondary Ewing sarcoma.
What about non Ewing round cell sarcoma?
Considering the very high percentage of pediatric patients affected by malignancies, 40 cases of EWS seems to be by chance rather than any connection.
That's probably why prognosis is the same as EWS as first tumor. For example, if a person had a previous car accident, the chance of having a second one is the same. He doesn't risk more the second time.
Therefore, the message is misleading.
We thank Reviewer 1 for his assessment. Because these are rare manifestations of a rare disease, we are aware of the limited power of the analysis. In particular, because Ewing sarcoma occurs in such low numbers as a second tumor, a purely incidental etiology is possible (and likely). Our data confirm that Ewing sarcoma does not appear to be an index tumor for tumor predisposition, nor is it regularly associated with pretreatment for the most common childhood cancers. For clarification, we have added the following sections in the Discussion:
“Because of the heterogeneity of treatment protocols for first-time malignancies, variable doses of chemotherapeutic agents, lack of clinical data on patients, and small EwS case series, the specific risk of these chemotherapeutic agents for EwS as SMN cannot be conclusively assessed.“
“There are several genetic syndromes with links to soft tissue and bone sarcomas, whereas EwS does not appear to be an associated index tumor, unlike osteosarcomas and embryonal rhabdomyosarcomas.“
“In conclusion, it remains unclear to what extent the type of treatment of the first malignancy or a genetic predisposition contributed to the development of the secondary EwS cases or whether these cases were incidental findings.“
By doing so, we believe that the message of the study becomes clearer.
Taking into account the limitations of a retrospective analysis, the data of the current study are based on an orderly statistical analysis from a long-standing database. Similar analyses have already been published in reputed journals. We ask the reviewer to specify his criticism of the design of the study.
In the database referred to, only patients with a diagnosis of Ewing sarcoma, i.e. after histopathological diagnosis, are explicitly included internationally. A data analysis with regard to non-Ewing round cell tumors is therefore not possible, since no data are available, but also not desired, since it is not the content of the paper.
Reviewer 2 Report
Minor comments:
1. Figure 1 is difficult to understand. please annotate differently.
2. The authors should consider enhancing the discussion section by making it clear to the reader that secondary Ewing sarcoma is not seen at a higher incidence than other secondary tumors.
3. In addition, please include available data to describe the incidence of other secondary malignancies not associated with constitutional genomic alterations such at Li-Fraumeni.
4. Are the authors able to include treatment responses for these secondary Ewing sarcomas identified? Were the outcomes similar to primary treatments?
Author Response
We thank Reviewer 2 for his constructive comments and suggestions for improvement.
Minor comments:
1. Figure 1 is difficult to understand. please annotate differently.
We have renewed Figure 1 and added the corresponding legend.
2. The authors should consider enhancing the discussion section by making it clear to the reader that secondary Ewing sarcoma is not seen at a higher incidence than other secondary tumors.
We have added the following passages to the discussion section:
“The incidence of EwS as a second malignancy is presumably lower than that of other solid bone and soft tissue tumors such as rhabdomyosarcomas and osteosarcomas.“
3. In addition, please include available data to describe the incidence of other secondary malignancies not associated with constitutional genomic alterations such at Li-Fraumeni.
We have added the following passage to the discussion section:
„There are several genetic syndromes with links to soft tissue and bone sarcomas, whereas EwS does not appear to be an associated index tumor, unlike osteosarcomas and embryonal rhabdomyosarcomas.“
4. Are the authors able to include treatment responses for these secondary Ewing sarcomas identified? Were the outcomes similar to primary treatments?
Following this comment, we have designed a new Table 2 (“Clinical features at diagnosis and response to therapy in comparison between Ewing sarcomas as first and second malignancies in the EICESS 92, Euro-E.W.I.N.G. 99, and Ewing 2008 trials“), wherein treatment response is included. This table also includes a comparison between Ewing sarcoma as a second malignancy and Ewing sarcoma as a first and sole tumor. The results from this analysis were carried over into the results section and discussion.
Reviewer 3 Report
This an important study from Europe that adds to the North American data on secondary Ewing sarcoma. Overall, these data should be published, but there are a number of issues with the manuscript that should be addressed before acceptance.
MAJOR COMMENTS:
1. The intro should include at least some information about what the prior studies on this topic (refs 13-16) found, including ref 25, which is only referenced in the discussion.
2. Treatment details for each of the studies (p 3, lines 115-143) could be more easily described in a table form.
3. The manuscript describes one patient with LFS. It would be helpful to know the extent that germline data was/was not available on the whole cohort to contextualize this piece of data. If no germline data were available for most patients, this should be explicitly stated.
4. Figure 1 is confusing - in appears as though only 22 Ewings cases are shown by location in the blue dots (there should be 44). It is also unclear if the yellow dots are cases already shown in the blue dots or not (are they being double counted?) and this could be clarified in the legend.
5. The discussion references several analyses that are not explicitly shown in the results, specifically, mean age (results show median age), tumor volumes, and alkylator agent (line 343-347 and 350-354). It would be helpful to include a table of summary data (including the quantitation of site of secondary Ewings and of patients with pulmonary versus other mets) for the whole cohort, plus whatever descriptive statistics and univariate analyses were performed using these data for greater clarity overall.
6. The claim in the discussion lines 304 to 305 about the outcome of secondary Ewings being determined by the general risk factors of Ewings is not supported with any data stated in the results beyond the localized versus metastatic curves. If this is what the authors meant, it should be stated more specifically.
7. Lines 355-356 state the authors could not identify any risk factors, but that analysis is not evident in the results section (beyond primary heme. vs solid tumor primary). In general, the analysis of the summary data is not transparent and should be more explicitly shown (see comments 5 and 6 above).
8. The discussion is generally confusing to read. In part, this is because new data are being introduced throughout the discussion (see comment 5 above). In addition, the content of each paragraph jumps around and the same points are repeated across paragraphs. This section should be edited for organization to identify the key points and make it more concise and easier for a reader to follow.
MINOR COMMENTS:
1. Page 2 Line 81-82 has a sentence fragment.
2. Methods (p 3, line 146) suggests the analysis included secondary and tertiary EwS patients but most of the rest of the manuscript just talks about secondary EwS and the results state "data analysis does not allow valid information on the frequency of EwS as third malignancies" (line 192). Please clarify what cases were included. If analysis is not valid for the tertiary cases, they should be excluded.
3. Table 1 legend is confusing with regard to the asterisks. Did the patient with 2 asterisks have Ewings as a fourth malignancy? If so, then the manuscript should clarify this instead of calling both cases "third malignancies." Again, it seems that these cases should be excluded (see above comment).
3. Figure 4 is incorrectly designated as Figure 3 in the text (line 251).
4. Fig 4 legend has a typo on line 258 ("malignancy malignancies").
5. Page 8, line 269 has a typo.
6. Page 9, 288-290, the wording is very confusing.
Author Response
This an important study from Europe that adds to the North American data on secondary Ewing sarcoma. Overall, these data should be published, but there are a number of issues with the manuscript that should be addressed before acceptance.
We would like to thank Reviewer 3 for his positive feedback and constructive suggestions for improvement. We are of the opinion that through this precise analysis the manuscript gains in comprehensibility, conciseness, but also in content.
MAJOR COMMENTS:
- The intro should include at least some information about what the prior studies on this topic (refs 13-16) found, including ref 25, which is only referenced in the discussion.
We have added the following passage to the introduction section:
“Despite its rare occurrence, patients with secondary EwS appear to have clinical differences and a worse prognosis than patients with primary EwS.“
- Treatment details for each of the studies (p 3, lines 115-143) could be more easily described in a table form.
In addition to the text comments, we have created a new Table 1 (“Development of risk stratification and chemotherapy for Ewing sarcoma (EwS) in different trials from the Cooperative Ewing Sarcoma Study (CESS) group “) with the information on risk stratification and chemotherapy regimens in each Ewing sarcoma trial.
- The manuscript describes one patient with LFS. It would be helpful to know the extent that germline data was/was not available on the whole cohort to contextualize this piece of data. If no germline data were available for most patients, this should be explicitly stated.
To clarify the issue of predisposition in relation to Ewing sarcoma, we added the following passages to the Discussion:
“There are several genetic syndromes with links to soft tissue and bone sarcomas, whereas EwS does not appear to be an associated index tumor, unlike osteosarcomas and embryonal rhabdomyosarcomas.“
“It must be taken into account that we did not have information on germline mutations in most patients with EwS as SMN. We tried to retrospectively examine the tumor samples available to us with regard to P53 expression, but the number of available tumor material as well as the interpretation did not allow any clear conclusions, so that this information was not explicitly included in the study. In conclusion, it remains unclear to what extent the type of treatment of the first malignancy or a genetic predisposition contributed to the development of the secondary EwS cases or whether these cases were incidental findings.“
- Figure 1 is confusing - in appears as though only 22 Ewings cases are shown by location in the blue dots (there should be 44). It is also unclear if the yellow dots are cases already shown in the blue dots or not (are they being double counted?) and this could be clarified in the legend.
We have renewed Figure 1 and added the corresponding legend.
- The discussion references several analyses that are not explicitly shown in the results, specifically, mean age (results show median age), tumor volumes, and alkylator agent (line 343-347 and 350-354). It would be helpful to include a table of summary data (including the quantitation of site of secondary Ewings and of patients with pulmonary versus other mets) for the whole cohort, plus whatever descriptive statistics and univariate analyses were performed using these data for greater clarity overall.
For a more detailed presentation of the results, we have newly created Table 2 (“Clinical features at diagnosis and response to therapy in comparison between Ewing sarcomas as first and second malignancies in the EICESS 92, Euro-E.W.I.N.G. 99, and Ewing 2008 trials“), in which data on clinical characteristics as well as on treatment response in Ewing sarcoma as a second malignancy and Ewing sarcoma as a first and sole tumor is included. The results from this analysis were carried over into the results section and discussion.
- The claim in the discussion lines 304 to 305 about the outcome of secondary Ewings being determined by the general risk factors of Ewings is not supported with any data stated in the results beyond the localized versus metastatic curves. If this is what the authors meant, it should be stated more specifically.
- Lines 355-356 state the authors could not identify any risk factors, but that analysis is not evident in the results section (beyond primary heme. vs solid tumor primary). In general, the analysis of the summary data is not transparent and should be more explicitly shown (see comments 5 and 6 above).
We modified several passages in the results section and added data to name the results more clearly. We created Table 2 to support clarity and presentation of results. In it, we described a comparison between EwS first and second tumors.
We modified the text passages addressed regarding the risks for EwS as a second tumor:
“Comparison between EwS as primary and EwS as secondary malignancy did not reveal statistically significant differences in clinical features at diagnosis and response to treatment after induction therapy, except for tumor location at diagnosis (P=.03; Table 2). We observed fewer secondary EwS in the lower extremities and more thoracic and extraosseous secondary EwS.“
“Statistically, due to the small number of cases, a comparison of EFS/OS between the different EwS studies is not meaningful, partly because localized patients are mixed with metastatic patients. A similar limitation would have been the analysis of other prognostic factors known for primary EwS, such as tumor volume and histopathologic response to therapy.“
“Our analysis showed that the initial tumor volume, and the frequency and distribution of distant metastases at the time of diagnosis in secondary EwS is similar to that in primary EwS (Table 2) [7]. The difference in tumor location in our study between primary and secondary EwS tumors must be critically questioned due to the low case number for secondary EwS cases.”
Another paragraph in the discussion was adjusted:
„Overall, survival rates in patients with primary and secondary EwS are comparable, although this observation may be influenced again by the small number of cases in the present study. In a large previous study of secondary EwS and PNET cases, these patients showed worse overall survival than primary EwS patients, and this was related to older age with comorbidity [17]. That we no longer saw this association may be due to the newer data and improved treatment options including improved supportive care. Our prognostic analysis suggests that the outcome of patients with secondary EwS is determined by the greatest risk factor of EwS, the spread of the disease at diagnosis, and not by the diagnosis of secondary malignancy per se; this is comparable to the results of previous studies [4, 17]. As the most important unfavorable prognostic factor in both primary and secondary EwS is the presence of distant metastases at the time of diagnosis [28, 29], staging remains crucial in the diagnosis of secondary EwS. 8. The discussion is generally confusing to read. In part, this is because new data are being introduced throughout the discussion (see comment 5 above). In addition, the content of each paragraph jumps around and the same points are repeated across paragraphs. This section should be edited for organization to identify the key points and make it more concise and easier for a reader to follow.”
We have restructured the discussion section and tried to be more concise. In addition, the methods section has been supplemented. We hope that this will make the classification of our data analysis clearer and more interpretable for the reader.
MINOR COMMENTS:
- Page 2 Line 81-82 has a sentence fragment.
The specified sentence has been corrected.
- Methods (p 3, line 146) suggests the analysis included secondary and tertiary EwS patients but most of the rest of the manuscript just talks about secondary EwS and the results state "data analysis does not allow valid information on the frequency of EwS as third malignancies" (line 192). Please clarify what cases were included. If analysis is not valid for the tertiary cases, they should be excluded.
- Table 1 legend is confusing with regard to the asterisks. Did the patient with 2 asterisks have Ewings as a fourth malignancy? If so, then the manuscript should clarify this instead of calling both cases "third malignancies." Again, it seems that these cases should be excluded (see above comment).
We have made the method section more precise: “Patients with EwS as third or fourth malignancy were not included in this study.”
We have deleted the said sentence from the results section.
Based on this evidence and to ensure the uniqueness of the study, we restructured the patient cohort and removed 2 patients with third- and fourth-malignancies from the data analysis. Accordingly, after recalculation of the statistics, the table and figures, as well as the content explanations in the methods and results sections, were modified.
- Figure 4 is incorrectly designated as Figure 3 in the text (line 251).
- Fig 4 legend has a typo on line 258 ("malignancy malignancies").
- Page 8, line 269 has a typo.
- Page 9, 288-290, the wording is very confusing.
These spelling errors and misnomers have been changed. The manuscript was finally searched again for errors and these were corrected.
Round 2
Reviewer 1 Report
Despite the attempt to ameliorate their paper, I still believe that the data do not support the message of this paper. It can be misleading